# Comparison of Single Cell Transcriptome Sequencing Methods: Of Mice and Men

**DOI:** 10.3390/genes14122226

**Published:** 2023-12-16

**Authors:** Bastian V. H. Hornung, Zakia Azmani, Alexander T. den Dekker, Edwin Oole, Zeliha Ozgur, Rutger W. W. Brouwer, Mirjam C. G. N. van den Hout, Wilfred F. J. van IJcken

**Affiliations:** 1Department of Cell Biology, Erasmus University Medical Center Rotterdam, Wytemaweg 80, 3015CN Rotterdam, The Netherlands; b.hornung@erasmusmc.nl (B.V.H.H.); m.vandenhout@erasmusmc.nl (M.C.G.N.v.d.H.); 2Genomics Core Facility, Erasmus University Medical Center Rotterdam, Wytemaweg 80, 3015CN Rotterdam, The Netherlands

**Keywords:** single cell sequencing, PlexWell, Smart-Seq3, 10X genomics, FLASH-seq, SORT-seq, VASA-seq, HIVE, transcriptomics, benchmarking

## Abstract

Single cell RNAseq has been a big leap in many areas of biology. Rather than investigating gene expression on a whole organism level, this technology enables scientists to get a detailed look at rare single cells or within their cell population of interest. The field is growing, and many new methods appear each year. We compared methods utilized in our core facility: Smart-seq3, PlexWell, FLASH-seq, VASA-seq, SORT-seq, 10X, Evercode, and HIVE. We characterized the equipment requirements for each method. We evaluated the performances of these methods based on detected features, transcriptome diversity, mitochondrial RNA abundance and multiplets, among others and benchmarked them against bulk RNA sequencing. Here, we show that bulk transcriptome detects more unique transcripts than any single cell method. While most methods are comparable in many regards, FLASH-seq and VASA-seq yielded the best metrics, e.g., in number of features. If no equipment for automation is available or many cells are desired, then HIVE or 10X yield good results. In general, more recently developed methods perform better. This also leads to the conclusion that older methods should be phased out, and that the development of single cell RNAseq methods is still progressing considerably.

## 1. Introduction

For more than 10 years, single cell RNA sequencing (scRNAseq) has been one of the main technologies to transform science [1,2,3]. It has become common to not only investigate tissue, but also to zoom in onto individual (rare) cell populations, to differentiate between cell populations, between specialized cells within them, or diverging responses within the same cell population [4,5]. While some of the first scRNAseq methods were complex with a myriad of manual steps (e.g., [6] and references within), the ongoing development has resulted in a large variety of commercial suppliers and kits, which are remarkably diverse in the number of cells required, their protocol complexity, and equipment requirements.

Continuous development has improved the accuracy, sensitivity, and throughput of scRNAseq methods, but also created a plethora of methods to choose from. There are marked differences between these methods and choosing the right one for each application can be challenging. As a genomics core facility, which routinely performs single cell sequencing and implements new methods, we would like to recommend to our customers the best method for each application. In addition, when considering which methods to recommend to customers or which new methods to implement, various metrics need to be evaluated.

While the costs of reagents might be the most apparent for the customer, the required technician’s hands-on time or technological considerations are no less important. With a look to the underlying technology, it needs to be considered how diverse the underlying cell population is, and if a low-throughput method with a 96- or 384-well plate might be sufficient, or if a bigger population with many thousands of cells might be necessary. Other factors such as the necessary sequencing depth, the possible detection of isoforms or the sequencing of non-polyA-tailed transcripts make method selection not trivial.

Here, we evaluate a multitude of methods for their performance across a range of different quality control parameters. We discuss their suitability to deliver reproducible single cell transcriptomics data. These results provide guidance both for individual researchers, consortia, and for core facilities.

## 2. Materials and Methods

Detailed information for all methods can be found in the Appendix A.

### 2.1. Cell Growth and Sorting

K562 is a human multiple melanoma cell line and was obtained for the ATCC (ATCC CCL-243). Cells were maintained at 37 °C under 5% CO_2_ in RPMI medium supplemented with 10% FBS and penicillin-streptomycin.

Mouse embryonic stem cells (mESCs) were cultured as previously described [7]. Mouse ESCs were cultured on 0.1% gelatin and FCS-coated plates in N2B27 medium supplemented with 1000 U/mL LIF, 3 µM CHIR99021 (Axon Medchem, Groningen, The Netherlands), and 2 µM PD0325901 (Merck, Amsterdam, The Netherlands). Cells were cultured at 5% CO_2_ and 37 °C, passaged 1:5 every 2–3 days by trituration of colonies to a single cell suspension using 0.05% trypsin-EDTA.

Cells were counted with a Countess II from Invitrogen/ThermoFisher Scientific (Waltham, MA, USA). K562 and mESC cells for the plate-based single cell transcriptome assays (Smart-seq3, PlexWell, FLASH-seq, SORT-seq, and VASA-seq) were sorted in a checkerboard pattern into 96- or 384-well plates using CellenOne X1 (Scienion, Berlin, Germany). Cells were sorted into 96- or 384-well plates containing different cell lysis media, or were thermally or enzymatically lysed after sorting, depending on the method (see Appendix A). Plates were sealed and frozen at −80 °C in the case that processing was not directly started.

### 2.2. Single Cell Transcriptome Assays

Eight different single cell RNAseq methods have been performed: PlexWell, Smart-seq3 [8,9], FLASH-seq [10,11], 10X Chromium [12], VASA-seq [13], SORT-seq [14,15], Evercode WT mini (a commercial version of Split-seq [16]), and HIVE. All methods are described in detail in the Appendix A.

In brief, for the 5 plate-based methods (Smart-seq3, PlexWell, FLASH-seq, SORT-seq, VASA-seq), cells were dispensed into a 96- or 384-well plate with a CellenOne instrument. Cells were lysed, cDNA was generated, and in some methods, the cDNA concentrations were quantified and further checked on a Bioanalyzer. The cDNA was tagmented and amplified by PCR to generate Illumina sequencing libraries. The amount of input cells specified in Appendix A takes empty wells and other controls into account, and therefore only includes dispensed K562 and mESC cells. As an example, the 384-well plates for VASA- and SORT-seq contained 8 empty controls; therefore, only 376 cells are listed.

For 10X, 1 million/mL K562 and 1 million/mL mESCs cells were mixed 1:1 in PBS and processed according to the manufacturer’s instructions in the Chromium Next GEM Single Cell 3′ protocol v 3.1. Single cell emulsions were generated on the Chromium Controller (10X Genomics, Leiden, The Netherlands), and 8250 cells were loaded to target a recovery of 5000 cells.

For HIVE and HIVE CLX, the manufacturer’s instructions from the HIVE scRNAseq v1 Processing Kit User Protocol (Honeycomb Biotechnologies, Waltham, MA, USA) were followed. A total of 15,000 cells were loaded, with the target to recover 6000 cells for the HIVE and 30,000 on the HIVE CLX to recover 11,000. We processed three hives, with K562, mESC, and 1:1 K562:mESC mixture. One additional HIVE and HIVE CLX with both K562 and mESC cells each were frozen (approximately 6 months and 6 weeks, respectively).

For the Split-seq library, the instructions in the Evercode WT Mini manual v 2.1.2 from August 2023 (Parse Biosciences, Seattle, WA, USA) were followed, with minor modifications (see Appendix A).

### 2.3. Bulk RNAseq

RNA was isolated with the RNeasy plus Micro kit by Qiagen, with 500,000 cells per sample. Bulk total RNA was prepared from triplicates of K562 cells and mESCs according to the Illumina TruSeq stranded mRNA protocol (Illumina, San Diego, CA, USA). A part of the workflow was automated with the Bravo automated liquid handling platform (Agilent Technologies Inc, Santa Clara, CA, USA).

### 2.4. Sequencing

The generated libraries were sequenced on Illumina systems, either single read or paired-end reads of 50 bp (Smart-seq3, PlexWell, FLASH-seq, Bulk) or paired end 26 and 60 bases (VASA-seq, SORT-seq) were generated. For 10X, the libraries were sequenced yielding paired-end reads of 28 and 90 bp, for HIVE, paired-end reads of 26 and 51 bp.

Appendix A describes in detail the read length and sequencing system for each library.

### 2.5. Data Analysis

In brief, the data were processed in pipelines, designed to be as similar as possible to the different methods, which were implemented in Snakemake v 6.11.0 [17] and used the same reference genome, a concatenated FASTA file of GRCh38 [18] and GRCm38 [19] included in the cellranger software refdata-gex-GRCh38-and-mm10-2020-A [12]. Differences in the pipelines are attributable to intrinsic features of the sequencing method, differences in paired-end status, UMI presence, barcode detection, and are mainly restricted to the parameters used in STAR [20]. All details can be found in the Appendix A. It was ensured that all pipelines run the same version of all included programs. All read files were trimmed with CutAdapt [21] for 3′ adapters with the ‘–a’ option. In the case of paired-end data, both reads were trimmed together, with the additional option ‘–A’ (3′ adapters for the second read). Reads were mapped with star v 2.7.9.a [20]. The conversion of SAM/BAM files and attainment of the related statistics was performed with SAMtools [22]. For most analyses, all samples were normalized to 20,000 read pairs per cell on average, except for the estimation of multiplets, non-detected genes, and sequencing saturation, which were performed on the full data.

The python StatsModel package v 0.14 [23] was used for regression calculations. The single cell count matrices were further analyzed in R 4.2.1 [24] with Seurat 4.3.0 [25]. Figures were generated in R 4.2.1 [24] or in Python3 with Matplotlib v 3.5.1 [26]. Figures were assembled and annotated with Inkscape v 1.2.1 [27].

## 3. Results

We benchmarked a multitude of single cell transcriptome assays. For a systematic comparison of the methods, we used two cell types. The human K562 cell line, which is a very homogeneous cancer cell line, and mouse embryonic stem cells (mESCs), which are native mouse cells. Bulk RNA sequencing data of both cell lines were generated as a ground truth to assess differences between single cell and bulk assays. The study design, workflows, and outcomes are depicted in Figure 1. For most assays, single cells were dispensed by CellenOne (in one case F.SIGHT) into microtiter plates, except for Evercode and HIVE where both cell types were mixed, and 10X genomics, where the cells were put in emulsion droplets by the Chromium Controller. Subsequently, single cell libraries were made according to the published protocol or the manufacturer’s instructions, and then sequenced. In the case of HIVE and HIVE CLX, one out of four in each batch were frozen (six months and six weeks, respectively), according to the manufacturer’s instructions. Sequencing data were normalized per protocol to an average of 20 k clusters per single cell and aligned to a combined mouse and human reference genome. After read counting, further analysis and visualizations have been created to show the performance of each technology and compare them to each other and the ground truth (Figure 1).

### 3.1. Single Cell RNAseq Requirements

Besides the scientific results described in the following sections, the technological requirements also need to be considered. All workflows utilized various equipment, which can be expensive and complicated to acquire (Figure 1). HIVE is an exception, as it is a self-contained workflow. The bulk RNAseq workflow does not require any automation either but was partially automated with a Bravo automated liquid handler. The 10X workflow requires the Chromium Controller from 10X as the only machine. All plate-based methods require a method for cell sorting, in this case, the CellenOne or alternatively FACS. These methods have the additional disadvantage of requiring many liquid dispensing and pipetting steps, which for some (SORT- and VASA-seq) were performed with a Nanodrop II pipetting robot and an Echo 525 robot liquid handler, and for others (Smart-seq3, PlexWell, FLASH-seq) with a Mantis liquid handler, I.DOT liquid handler, and a Mosquito pipetting robot. In theory, these steps can be performed manually, but even with automation, these methods take 3–4 days to complete.

The overall required time between the different methods is quite comparable. Hands-on time ranges between 8 and 16 h, yet the total time, including incubation times and other logistic considerations like safe stopping points for freezing, is about 3 days. Bulk RNA sequencing with automation only requires 2 days, whereas VASA-seq needs 4 days. Despite similar levels of automation and commercial solutions being available, the hands-on time varies and ranges between 8 h for 10X, 9 h for SORT-seq, and up to 16 h for PlexWell and VASA-seq.

### 3.2. Quality Control

First, the alignment percentages of the two included reference genomes, the human GRCh38 and the mouse mm10 (Figure 2), were assessed. A 90% mapping ratio of reads (or of UMIs, where applicable) to one of the organisms was used to assign a cell to being either human or mouse, whereas cells with a lower percentage were assigned as a multiplet. Three of the five plate-based assays did not yield any mixed cells as expected due to the single cell dispensation by CellenOne. The 10X genomics, HIVE, HIVE CLX, Evercode, SORT-seq, and VASA-seq datasets indicated the presence of mixed cells, ranging from 2% (10X) to 9% (HIVE/HIVE CLX), with the Evercode WT Mini being an outlier (49%; full details are available in Appendix A). In the 10X, HIVE, HIVE CLX, and Evercode datasets, mixed cells also contained more overall features and a higher diversity. This is indicative of mouse–human cell duplets in a single droplet/well as it is inherent to the methodologies. Remarkably, in the SORT-seq and VASA-seq datasets, mixed cells were called, despite the single cell dispense by CellenOne. All parameters for mixed cells were in the same range as for the individual single cells. Most of the mixed cells in these datasets remained close to the 90% cutoff.

A high amount of mitochondrial RNA is indicative for a poor cell condition, and it is recommended to remove those cells from downstream analyses. The percentage of mtRNA was plotted for each assay (Figure 3; detailed numbers are available in Appendix A).

On average, the percentage of mitochondrial RNA is higher for human K562 cells than for mouse ESCs. Bulk sequencing resulted in the least amount of mitochondrial RNA in human cells, but in mouse cells, Evercode, HIVE CLX, and VASA-seq resulted in less mtRNA (bulk 2.24%, Evercode 0.7%, HIVE CLX 1.3%, VASA-seq 1.4%). These methods also showed the least amount of mtRNA in human cells, but more than bulk sequencing (bulk 1.8%; least amount in Evercode, 2.4%). A minimum amount of mtRNA is physiological for all cells, as no cell had 0% of mtRNA, but should still be minimized for information gain.

Single cell handling and dispense adds additional stress to cells which translates to higher mtRNA percentages. Smart-seq3 and SORT-seq show an elevated percentage of mtRNA compared to the other plate-based methods, SORT-seq especially in mouse cells. This shows variation between methods: The plates for both VASA- and SORT-seq were prepared at the same time, yet VASA-seq resulted in better values for the mtRNA. In mouse cells, the amount of mtRNA remained below 5%, except for SORT-seq with an average of 12%, whereas in human cells, it remained mostly below 10% (except for SORT-seq and Smart-seq3 body with 14%, and 10X nearly reaching 10%) [28]. Most methods maintain the mtRNA at over 90% of the human cells below the cutoff (with the exception of 10X, SORT-seq, and Smart-seq3 body), and in more than 80% below the cutoff for mouse cells (with the exception of SORT-seq and Smart-seq3 body). No bimodal distribution was detected in the sequencing data which had a higher average of mtRNA (Smart-seq3, SORT-seq, and 10X). Despite these good averages, filtering remains necessary, since some cells exceed the average by far. The highest amount of mouse mtRNA in a single cell was recovered in Smart-seq3 body (32%) and in humans in the 10X data (87%), and such cells should be excluded from further processing.

### 3.3. Performance of the Single Cell Methods

An important metric to assess is the complexity of the library: are there many or only a few different transcripts captured? The Shannon index [29] is a metric to evaluate diversity, which was used to evaluate the spread of coverage over the various genes. A zero indicates that all data points are equal (e.g., all read counts are equal to 1), while higher numbers indicate a more unique data spread. Bulk RNAseq captures the most transcripts, resulting in a diversity of 8.5 for both human and mouse. For the single cell sequencing methods, HIVE CLX resulted in the most diverse read mapping for both cell types (7.4–7.5), closely followed by PlexWell, VASA-seq, HIVE, Evercode, and FLASH-seq in humans (7.2 or better), and closely followed by PlexWell, VASA-seq, and FLASH-seq in mice (7.2 or better). The overall range of values was not spread quite far, with the worst values obtained by Smart-seq3 with values of 6.6–6.7 (Figure 4). Mixed cells from the 10X, HIVE/CLX, and Evercode data resulted in a higher diversity than single-species sequence sets, but this was not observed for the mixed cells from VASA- and SORT-seq (not shown). Smart-seq 3 showed the lowest diversity both for body and UMI reads in human, and lowest and third lowest in mouse.

Next, the number of features detected per assay on the normalized data was examined (Figure 5). The average of detected genes in the single cell assays approximated mostly around 2000–4000 (precise numbers can be found in Appendix A). Smart-seq3 UMI had the lowest averages (2400 human, 1600 mouse), and HIVE/CLX, PlexWell and FLASH-seq detected the most features in both cell types. For the K562 data, FLASH-seq, PlexWell, HIVE/CLX, and 10X are comparable, whereas in the mouse cells, 10X did not perform. The highest number of features in a human single cell was detected by Evercode and HIVE CLX with approximately 8400, and in mouse by Evercode with 8400.

We further investigated the feature overlap between the different single cell technologies and bulk RNA sequencing. Most features, which were detected in bulk sequencing, were also detected in HIVE CLX, with ~800 human and ~500 mouse features not detected in HIVE CLX, in both cases followed by HIVE and Evercode (Figure 6). Of all the features which were detected between at least one single cell method and the bulk RNAseq (14,141 to 20,030 in human cells, and 13,085 to 18,601 in mouse cells), most were detected consistently between all of the investigated methods (11,865 features in human cells, and 11,066 features in mouse cells). The lowest number of features from the bulk sequencing was detected in PlexWell for human (~6700 not detected) and for SORT-seq in mouse (~6000 not detected). Surprisingly, the single cell sequencing methods also detected a range of features not detected in bulk sequencing. The HIVE CLX technology detected the most, with more than 6000 in human cells and more than 4800 in mouse cells. The lowest number of extra features was detected by PlexWell and FLASH-seq in humans (~500) and mouse by SORT-seq (~300). It was further investigated how the features, which were not detected in the various single cell methods, were ranked in the bulk sequencing data, e.g., if non-detected genes were highly or lowly expressed. For both human and mouse, more than 50% of non-detected genes in the single cell methods ranked in the lowest 25% of expressed genes in bulk, and more than 75% in the lowest 40% of expressed genes in bulk (if one outlier is excluded 93% and 92% on average are in the lowest 40% for human and mouse respectively), indicating that in the single cell methods mostly lowly expressed genes are missed. All single cell methods (except for one sample of HIVE CLX in mouse) missed at least one gene in the top 50% of expressed genes, with some methods even missing genes in the top 2% of expressed genes. In mouse cells, all methods except for the HIVE also missed genes in the top 20% of expressed genes. This number was slightly lower in human cells. In general, the more overlapping genes are detected between a single cell method and bulk sequencing, the less likely it is that a highly expressed gene was not detected.

In contrast, it was also considered how many genes were detected in the single cell methods, but non detected in bulk sequencing. Also here, most genes which were newly detected ranked rather low in expression, with on average more than 80% of newly detected genes ranking in the lowest 20% of expressed genes, and on average more than 97% in the lowest 40% of expressed genes. Not all newly detected genes ranked lowly, with some methods also detecting new genes with a high gene expression, up to the top 10% or even top 5% of expressed genes.

Besides the total number of features, the relationship between new features gained and additional sequencing depth is relevant. Therefore, the ratio between these two variables on the non-normalized data was investigated (Figure 7). In humans, the HIVE CLX yields the best ratio of features to reads, followed by Evercode and 10X. In mice, the first two places are swapped, with Evercode yielding the best ratio, followed by HIVE CLX and 10X. The biggest difference can be seen for SORT-seq and HIVE. For SORT-seq, it does not perform well in the K562 cells together with Smart-seq3 body, but performs well in mouse cells. In contrast, for HIVE, it performs well in human cells, but has the worst yield in mouse cells. Sequencing saturation is not reached at 200,000 reads per cell for most methods, except for FLASH-seq and PlexWell in mouse, where the saturation plateaus after this point.

In the case of methods where the cell is assigned based on a barcode, rather than an Illumina index, the barcoding efficiency needs to be factored in. The sequencing data need to be separated into the distinct barcodes, and not all barcodes will be derived from cells, but rather from background. For the data presented here, SORT-seq, VASA-seq, and 10X Next GEM 3′ had the best efficiency, as 85–92% of the data were assigned to cells. This efficiency was lower for the HIVE and HIVE CLX, where only 60% of the data were assigned to a cell. The Evercode WT method showed a difference between human and mouse cells, as in human cells 59% of the data were retained, whereas in mouse cells only 38% were retained, and the combined libraries were placed in between them (50%).

One of the differentiating characteristics of single cell assays is full or partial transcript coverage. Therefore, the distribution of reads over the whole gene was investigated and shown in Figure 8. Each assay shows its own transcript coverage profile, which is always as expected from the library construction technology used. Bulk RNAseq and FLASH-seq yielded the most equal coverage, whereas a bias for an increased coverage at either the 3′ or 5′ was visible in most other methods. The plate-based assays (Smart-seq3 body, FLASH-seq, and PlexWell) have coverage over the whole transcript, and from both strands of the genome due to the paired-end sequencing. 10X, Hive, VASA-seq, Evercode WT mini, and SORT-seq yielded only reads from the sense strand over the whole length of the transcript, and show a preference for 3′ end reads. Some of the more internal reads have previously been attributed to semi-random binding of internal polyA repeats [30]. UMIs of the Smart-seq3 protocol were only detected on the 5′ end of the genes. To quantify the imbalance in coverage from 5′ to 3′ end of the transcript, we calculated the relative coverage per exon (reads/base) over each gene. For all genes, which had at least half of their exons covered, the relative standard deviation over their exon coverage was calculated and averaged per dataset (Appendix A). FLASH-seq shows the least imbalance with 14.9% relative standard deviation, followed by bulk RNAseq with 15.8%, and then by PlexWell with 16.1%, Smart-seq3 body with 17.2%, Evercode WT with 18.3%, and VASA-seq with 19.1%. The methods with known 3′ and 5′ bias had clearly higher deviations, with 10X having 20.1%, HIVE and HIVE CLX having 21.1%, Smart-seq3 UMI with 24.1%, and SORT-seq with 24.7%.

### 3.4. Comparability of Profiles

One of the main questions is how comparable and reproducible the transcript profiles are between any of these methodologies. UMAP grouping with Seurat showed on the first component a separation of cells into human and mouse (Figure 9A). Furthermore, three other main observations can be made from this plot. The first one is that the VASA-seq data only slightly cluster with the other methods, as can be seen for the human cells in Figure 9B. The second observation is that the bulk RNAseq data group within the VASA-seq cluster. The third observation is that for the mouse cells the grouping is based on the method although all technologies group together in lower dimensions, except for VASA-seq. The UMI and body components of Smart-seq3 are also grouped together but show a clear separation into both components. The PlexWell and FLASH-seq methods are derived from Smart-seq2, and group together here. If the technologies are investigated separately, then also a batch effect is visible for the mouse cells, but not the human cells. Otherwise, most of the mixed cells from both 10X and HIVE are forming separate groups, which cannot be seen for the SORT- and VASA-seq mixed cells. Overall, all methods are consistent, and show good agreement and reproducibility. The technological impact of all methods is less than the biological impact of the used cell material. This can also be seen in Figure 10, which is another representation of all the combined datasets. In this case, we correlated gene expression between the datasets (each dataset treated as a single expression profile). The correlation within one method is in general high, exceeding 0.8 and in most cases 0.9. Most datasets from different methods show a moderate correlation of 0.7 or higher to other datasets, with the exceptions of VASA-seq, which shows mostly a different profile, and Evercode WT, which shows a clearly distinct profile. The correlation of bulk RNAseq to other datasets did not differ considerably from differences within the single cell methods (except for Evercode), giving no method a noticeable advantage over others.

## 4. Discussion

The field of single cell sequencing is growing in complexity and new methods appear every year. This development has resulted in many commercial suppliers and kits, which are remarkably diverse in the number of cells required, their protocol complexity, and equipment requirements. We compared multiple available methods to evaluate their advantages and disadvantages and to provide guidance for individual researchers, consortia, and core facilities.

### 4.1. Time Requirements and Automation

Most of the methods can be (partially) automated, to save hands-on time and reduce errors. The plate-based assays require liquid handling and pipetting robots for efficient use. This can be a big obstacle if these are not already present in a laboratory, since purchase costs can be prohibitive. If they are available in a laboratory, then their use can make any of the described assays efficient, with absolute handling times of less than 4 days (including waiting times), and reduced error prone manual steps. Without any robots, variable handling and incubation times for a larger number of cells would negatively impact the results. There are two main alternatives, if such robots are not available. The first is the 10X platform, which requires only one machine and has everything necessary for single cell preparation built in. This decreases the complexity of the preparation, but increases the upfront capital cost and the throughput. The second alternative are methods which do not require any equipment and as much upfront capital investments, such as the HIVE or Evercode. Also, here the costs increase, due to the commercially supplied package, but are less of an investment than the 10X instrument. Since the necessary materials are disposable, they need to be bought for each preparation. Such methods are therefore the most suitable for laboratories, which do not have any instruments available nor perform single cell sequencing regularly.

The required hands-on time after automation is less of a decisive factor than generally anticipated. A total hands-on time of 8 h for 10X is a big relative time difference to the maximum of 19 h hands-on time for Evercode. However, the number of laboratories with sufficient throughput to make this a deciding factor is relatively small. The decision between a low- or high-throughput method is more likely to make a difference.

### 4.2. Filtering Cells

Apoptotic cells will be depleted of chromosomal RNA, due to the loss of membrane integrity; therefore, apoptotic cells will mostly yield the remaining mitochondrial RNA. It has been best practice to date to remove apoptotic cells with a mitochondrial RNA percentage higher than 5%, due to earlier indications of this being a reasonable threshold [31]. A recent publication reported this as 10% for human cells and 5% for mouse cells [28] though, and in some protocols, 15% is used [32]. The difference between human and mouse cells is also visible in the data presented here. For all methods, except for SORT-seq, the fraction of mitochondrial RNA was higher in human cells than in mouse cells. A non-negligible part of human cells also exceeded the 10% threshold (Smart-seq3 and 10X), although only in the Smart-seq3 datasets the 15% threshold is exceeded by a considerable number of cells. Contrary to human, in mouse cells the 10% threshold is rarely exceeded (except for SORT-seq). Overall, this correlates with the observations by Osorio et al. [28]. It would be sensible to derive a threshold per method or per dataset, but in all our datasets, no bimodal distribution is visible; therefore, it is not possible to derive a binary state of being apoptotic or not and to filter on this, based purely on the mtRNA amount.

### 4.3. Multiplets

The intermixing of human and mouse cells facilitates estimation of the rate of multiplets in a dataset. Other research showed multiplet rates from 2.5 to 37% [33] and 10X predicts a multiplet rate between 0.8 and 8% depending on the cell number [34] (although this has been reported to be higher [35]). With a mixing of two different species, we assume that we are able to detect 50% of all multiplets, as we will detect human/mouse and mouse/human multiplets, but not mouse/mouse or human/human multiplets. In our 10X data of 2500 cells, we detected 2% mouse/human multiplets. Taking into account the non-detectable, 1% human/human multiplets and 1% mouse/mouse multiplets, results in a 4% multiplet rate. This is higher than the predicted multiplet rate by 10X (below 2.5%). The same holds for HIVE, where the predicted multiplet rate is 9% for our HIVE datasets and 14% for the HIVE CLX datasets [36], and our data show an inferred multiplet rate of 18%. An outlier in this case is the Evercode data, as the multiplet rate was significantly higher than expected (49% detected, versus theoretically less than 2% [37]). It was however noted by the manufacturer that mixing of cells with uneven RNA content can lead to failure of the cells with lower RNA content, as they will be underrepresented in contrast to the cells with higher RNA content. A lower amount of sequencing output from the mESC cells in the pure libraries was noted, and therefore this can be a contributing factor, as a good part of the multiplets might be genuine mESC cells with low RNA content, mixed with high K562 background.

As shown in Figure 2, cell multiplets of distinct species are easily detected, due to the varying mapping rate of these cells, the elevated feature rate, and increased Shannon diversity. Difficult to detect are multiplets of same-species cells. As can be seen from the data in this manuscript, such cells are not detectable by simple metrics. While the multi-species multiplets show distinct characteristics compared to the single cells, this is due to the greater genetic diversity within the double-species multiplets, and therefore not applicable to single-species multiplets. A simple filtering for the cells ranking highest for features or diversity is not applicable either, since the double-species multiplets may rank lower than some of the single-species cells in these comparisons. It also cannot be excluded that the highest scoring single-species cells are single-species multiplets. Various computational methods have been developed [33] to detect these cell multiplets, but these were not benchmarked, as this was not the focus of this research.

Only a few multiplets were seen in the plate-based assays, as expected. Some cells in the SORT-seq and VASA-seq samples showed a human/mouse (or mouse/human) ratio below 90%, which is in principle not expected in a plate-based assay (although more than half of these showed a ratio >85%), but before sequencing all samples are mixed; therefore, barcode hopping, background RNA, etc. could be wrongly assigned. These “multiplets” did not show any characteristics of real multiplets, e.g., an elevated number of features or a higher diversity, and are therefore probably misidentified. The other plate-based assays showed no mixed cells, as expected.

### 4.4. Batch Effect

Multiple batches of K562 and mouse embryonic stem cells were used in this benchmarking study. Two different main observations are made in this regard. The first observation is that the results for the K562 cell line and the mESC can be clearly distinguished. Mouse and human cells were separated by both tSNE and UMAP and showed different profiles. But while the clustering of the K562 cells yielded mostly overlapping groups for the different methods, the mESC cells showed more of a gradient between the different methods. The K562 cell line is a stable human cancer cell line, and in theory, no major differences would be expected. In contrast, the mESCs do not seem as a biologically defined group of cells, and vary, potentially due to biology and due to the amount of passaging in the laboratory. Despite this, a clear grouping was still visible for both cell types.

### 4.5. Overlap and Difference between the Methods

The second main observation is that most methods overlap to some degree. VASA-seq is an exception, showing a strong separation from the other methods, likely due to the inclusion of more unique transcripts and non-polyadenylated transcripts, amongst which are histone genes. The bulk RNAseq was grouped within the VASA-seq data, which on first glance could be attributed to the additionally captured transcripts from VASA-seq, yet the overlap between the detected genes shows that this cannot be the sole cause, as other methods detect more overlapping transcripts. Some of the additionally detected histone genes show a high expression; therefore, the cause is more likely quantitative, in addition to being qualitative. Three of the investigated methods are in principle similar and derived from each other (Smart-seq3, PlexWell, and FLASH-seq being variations/further developments of Smart-seq2), which is also visible in the results. SORT-seq and 10X are both 3′ methods, but despite the principal similarities in technology, do not form a strong overlapping cluster. Despite these general differences between the methods, the agreement is in general high, which indicates that most results probably can be trusted independent of the method, but not yet high enough that a mixing of different methods within one experiment can be recommended.

## 5. Conclusions

To conclude, multiple single cell sequencing methods that vary widely in methodology were compared to each other and to bulk RNAseq (see Figure 11). In the case that researchers do not need the single cell resolution, we would advise the use of the bulk RNAseq method, since for many metrics, the bulk outperforms single cell methods. From the tested single cell RNAseq methods, Smart-seq3, which is the oldest full-transcript method used in this investigation, shows sub-par results, and we recommend researchers to search for better performing methods. The metrics of the 10X data also do not compare favorably in terms of transcript coverage and multiplets, but 10X still has the advantage of yielding the highest throughput, which the other methods do not offer, except for HIVE and Evercode. The HIVE seems to be the most suitable for laboratories, which do not have the necessary equipment for the other methods and require a high throughput. The Evercode method has in principle the same advantage, but cannot be recommended due to the issues with the multiplets, which makes it not suitable in many situations, and use cases will require more background knowledge. Moreover, with HIVE, samples can be stored before performing the scRNA library preparation, which allows for sample retrieval over time before processing or sending core facilities to sequencing in one batch. As shown in the results, there are no notable differences observed, making storage indeed a viable option. VASA-seq shows good results and detects non-polyadenylated transcripts, which other methods do not. VASA-seq could therefore be the method of choice, especially if non-polyA transcripts are of interest. FLASH-seq and PlexWell show comparable performances in many aspects and can be good alternatives if non-polyadenylated RNAs are not of interest. However, the PlexWell kit has been recently discontinued by the manufacturer, and therefore cannot be recommended anymore.

All methods have their advantages and disadvantages, which need to be evaluated based on the biological background and proposed application, and a global single recommendation is hard to formulate (Figure 11); therefore, we propose multiple. (1) For laboratories with the necessary robotic equipment, VASA-seq and FLASH-seq show good performance. For laboratories without robotic equipment, which plan to perform any single cell experiment, it depends on the throughput; therefore, three scenarios can be distinguished. (2A) For many experiments which require high cell counts, 10X would be recommended. (2B) For many experiments which require low cell counts, the acquisition of the necessary robots to perform VASA- or FLASH-seq can be recommended. (2C) For occasional single cell experiments, HIVE can be recommended.

## Figures and Tables

**Figure 1 genes-14-02226-f001:**
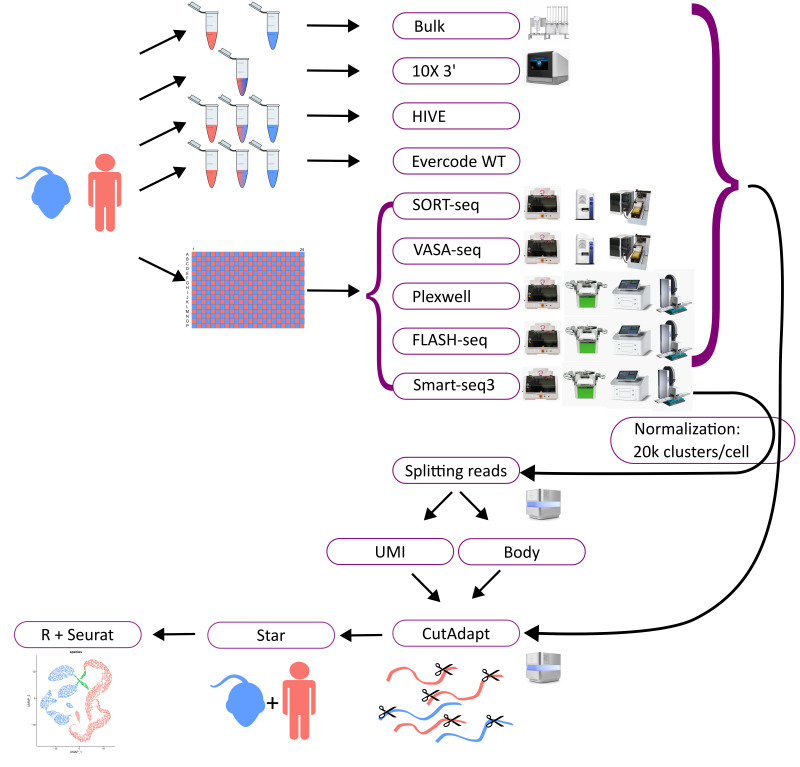
Overview methods: the diagram depicts the workflow. Mouse and human cells were utilized for all methods. Cells were applied separately or mixed for some of the workflows (bulk RNAseq, 10X, HIVE), or were sorted into a well plate (96 or 384) in a checker-board pattern, alternating human (red) and mouse (blue) cells (for Smart-seq3, PlexWell, FLASH-seq, VASA-seq, and SORT-seq). All workflows utilized various equipment, depicted next to the workflows, except for HIVE, which is a self-contained workflow. All samples were afterwards sequenced on an Illumina sequencer and normalized to 20,000 reads per cell on average. The data were then trimmed with CutAdapt, mapped with Star, and analyzed with Seurat. For Smart-seq3, the UMI and body reads were divided and analyzed separately.

**Figure 2 genes-14-02226-f002:**
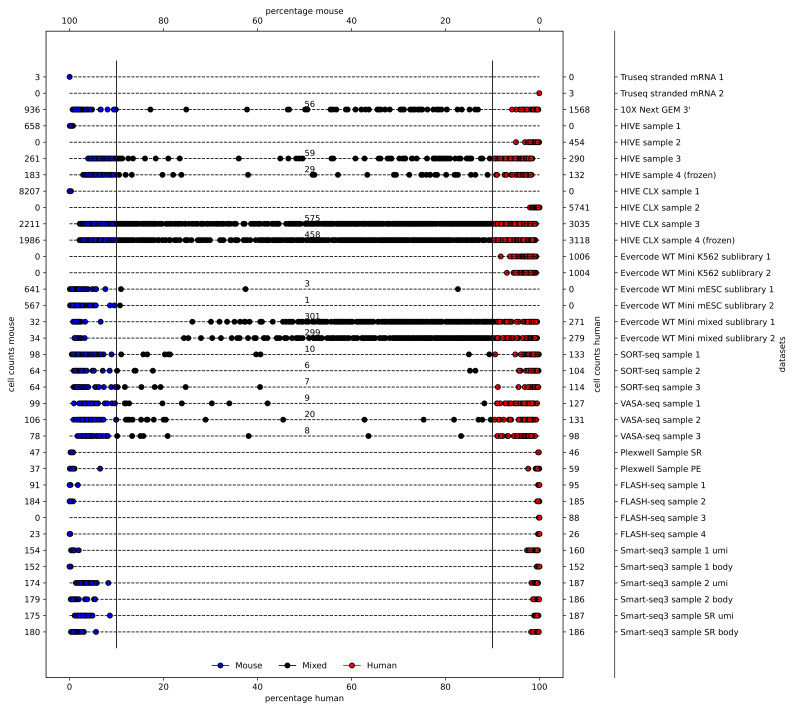
Multiplets: a varying number of multiplets is expected per method. A species cutoff of 90% is used to define mouse (blue), human (red), or mixed cells (black). Alignment percentages of mouse and human are indicated at the top and bottom, respectively. The numbers of counted cells for mouse and human are indicated on the left and right, respectively.

**Figure 3 genes-14-02226-f003:**
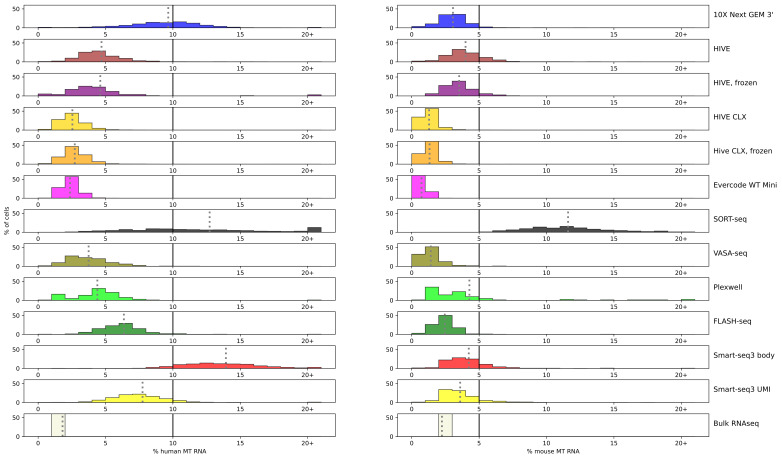
mtRNA: percentage of mtRNA is shown for each technology marked by color and plotted against the percentage of cells. Each bar represents a range of 1% point, except for the last bar after 20+, which accumulates all cells with a mtRNA percentage of more than 20%. In each subplot, the cutoff of 10% for human and 5% for mouse cells is indicated with a black line. A grey dotted line represents the average for the technology.

**Figure 4 genes-14-02226-f004:**
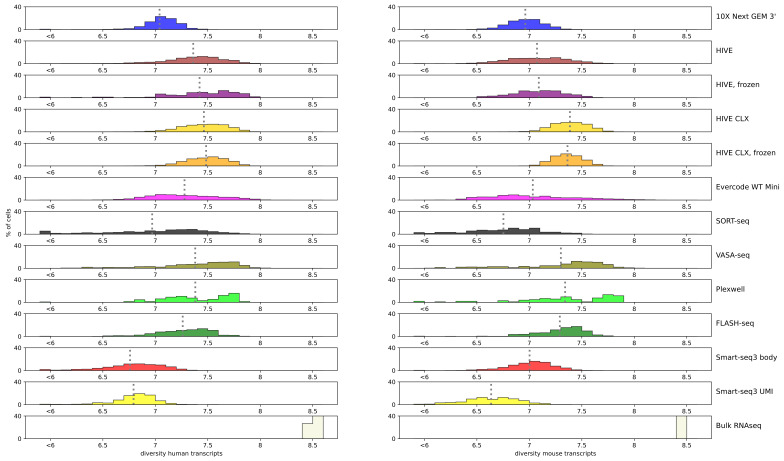
Diversity: Shannon diversity value per technology is indicated by the color plotted against the percentage of cells. Each bar represents a range of 0.1, except for the bar below 6, which accumulates all values below 6, independent of the actual value. A grey dotted line represents the average of the technology.

**Figure 5 genes-14-02226-f005:**
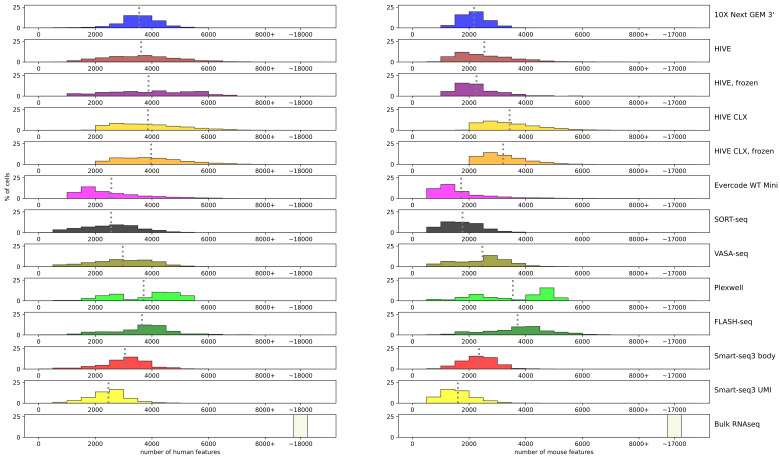
Features: number of features detected per cell. Each bar represents a range of 500 features, with two exceptions. The bar labeled 8000+ represents all cells with more than 8000 features, independent of the actual value. The bar labeled ~17,000/~18,000 contains the three replicates for bulk RNAseq data, which contains these high number of features. A grey dotted line represents the average of the technology.

**Figure 6 genes-14-02226-f006:**
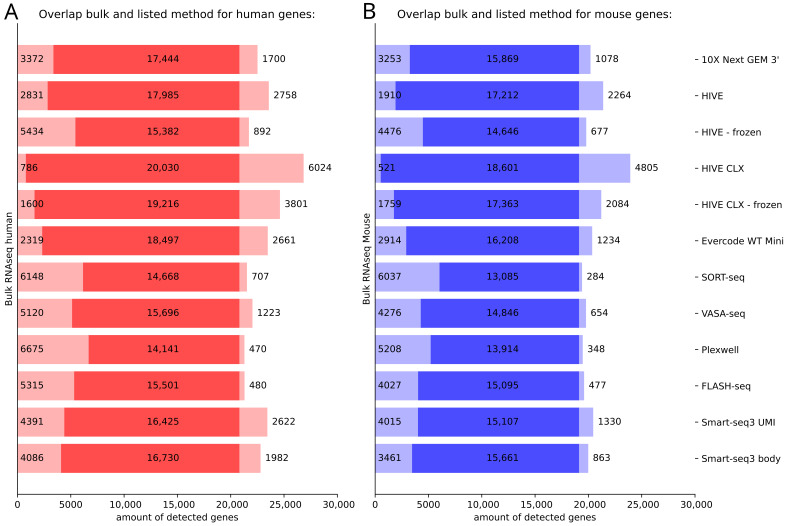
Detection of genes in single cell RNA sequencing methods in comparison to bulk RNAseq. (**A**) Human data, (**B**) mouse data. In each subpanel, on the left, the expression of genes only detected in bulk RNAseq is depicted, on the right, the expression of genes only detected in the various single cell methods (for cells identified as human/mouse in the subpanel (**A**,**B**). The intensely colored middle of the bar represents the genes which were detected with at least one read in both methods.

**Figure 7 genes-14-02226-f007:**
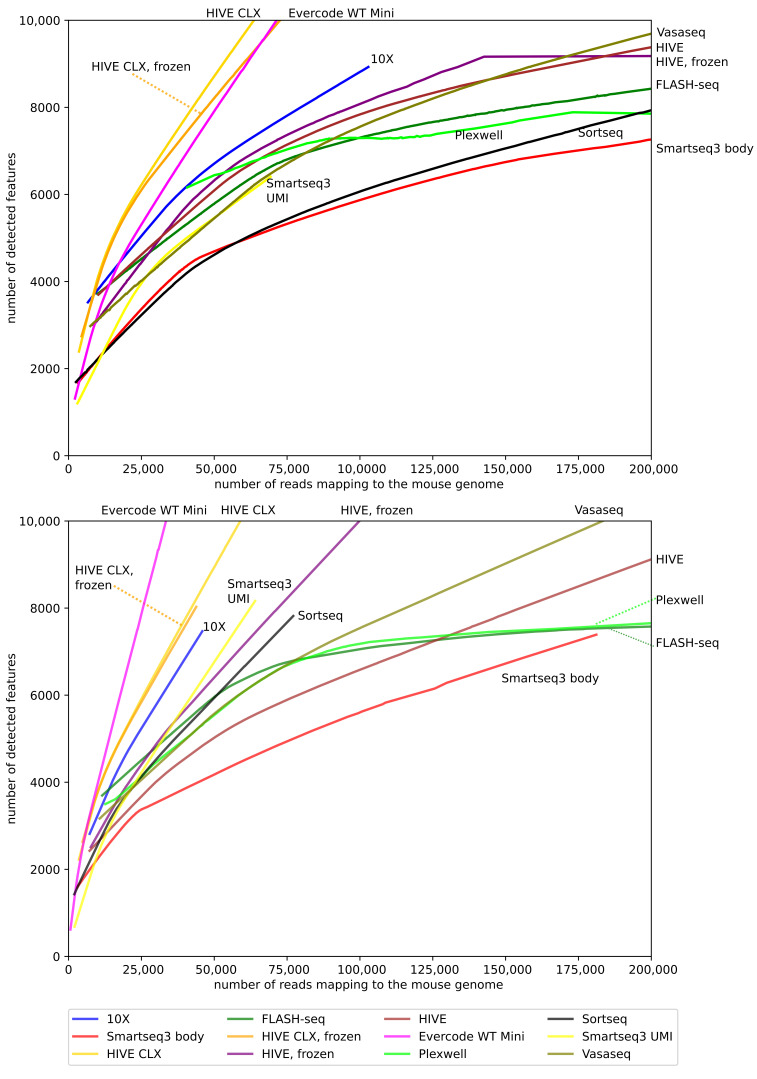
Reads to features: this diagram shows the identified features per cell versus the number of reads for that cell. Only cells with a maximum of 200,000 reads are displayed, as only a minor number of cells contained more reads.

**Figure 8 genes-14-02226-f008:**
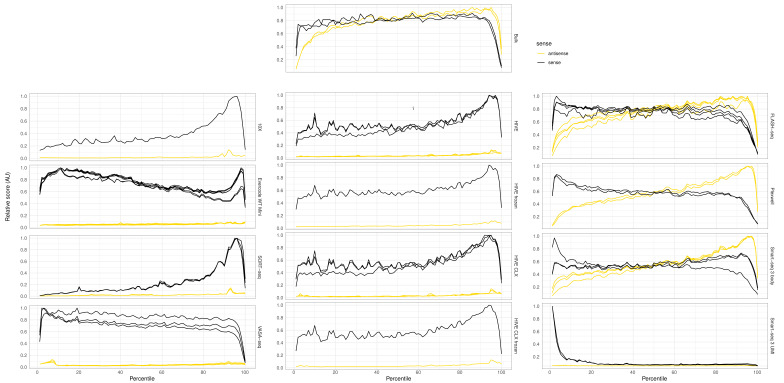
Gene coverage: relative coverage over features. The *x*-axis shows the relative length of a transcript, from 5′ to 3′. Black lines indicate coverage from reads aligning to the sense strand of the genome, yellow lines indicate coverage from reads aligning anti-sense, e.g., for Smart-seq3, the UMI is only present in reads starting from the TSO at the 5′ end of the transcript. Therefore, in Smart-seq3 UMI, we see high coverage at the 5′ side of the transcripts, and only alignment on the sense strand of the genome. In contrast, Smart-seq3 reads body are derived from paired-end sequenced, tagmented full-length cDNA, yielding reads over the whole transcript and on both strands of the genome.

**Figure 9 genes-14-02226-f009:**
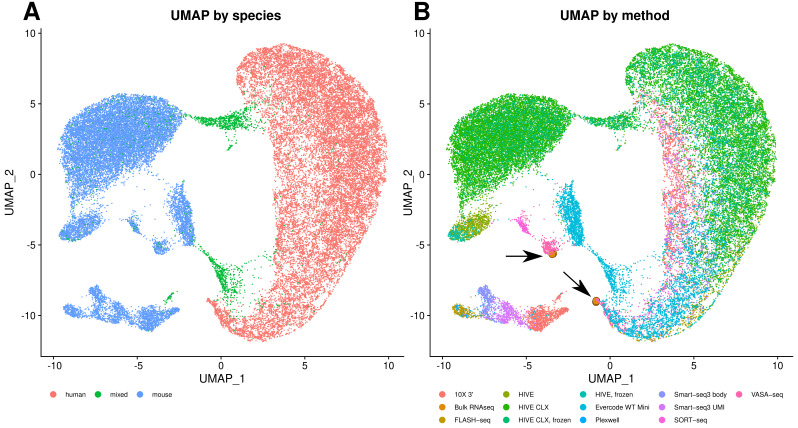
Cell clustering. (**A**) UMAP of all combined datasets, color based on species. (**B**) UMAP of all combined datasets, color based on method. The bulk RNAseq points have been enlarged and highlighted with arrows for better visibility. Both diagrams were generated with eight PCA dimensions.

**Figure 10 genes-14-02226-f010:**
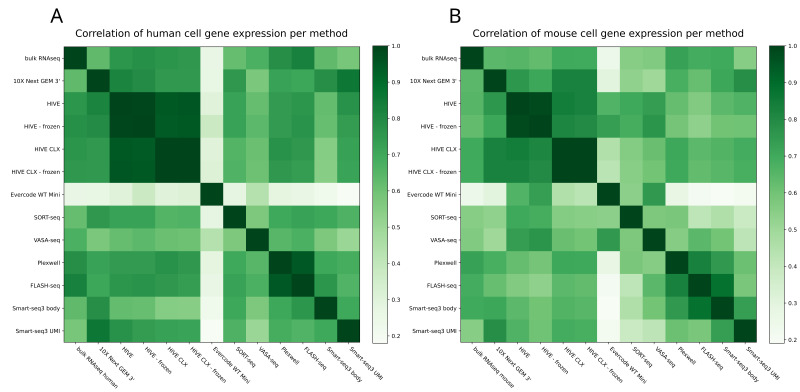
Correlation matrix: the Pearson correlation of all expression profiles. (**A**) Correlation of human gene expression. (**B**) Correlation of mouse gene expression.

**Figure 11 genes-14-02226-f011:**
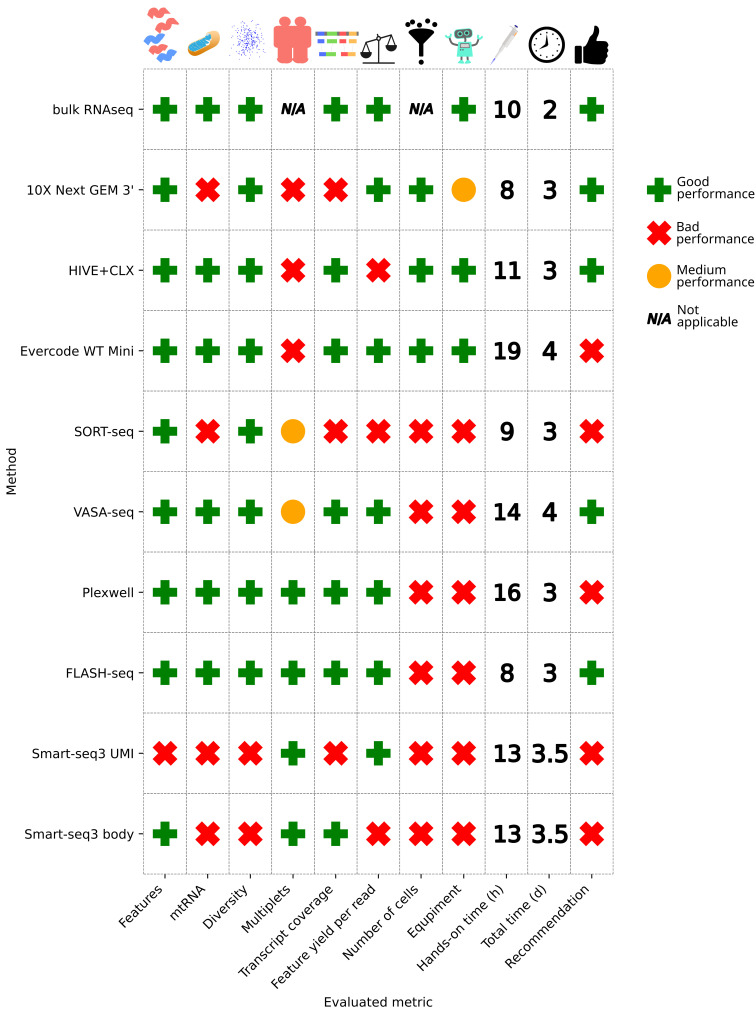
Overview of performance metrics. A method failed on a metric, if it showed worse performance in comparison to the average of the other methods or did not meet known standards (e.g., mtRNA cutoffs). If two groups were apparent, then the worse performing group was marked as failed, the better as passed. SORT- and VASA-seq achieved a medium score for the multiplets, since in theory, no multiplets should be present, yet we still detected some. 10X gets a medium score on the equipment requirements, since only one machine is necessary, in contrast to no equipment required or multiple robots being required. A recommendation for a method is given if it passed at least half of the evaluated criteria. The exceptions are PlexWell, which cannot be recommended anymore since it has been discontinued (although a new kit is available), and Evercode, since the multiplet issue makes it not suitable in many circumstances.

## Data Availability

All data have been uploaded to the EBI under projects PRJEB67541, PRJEB67543-PRJEB67549, and PRJEB67943.

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
