# Peer review of "Comparison of Single Cell Transcriptome Sequencing Methods: Of Mice and Men"

_genes, 2023, doi:10.3390/genes14122226_

Round 1

Reviewer 1 Report

Comments and Suggestions for Authors

SUMMARY

Here the Authors compared several scRNA-seq methods to evaluate their adantages and disadvantages and to provide guidance for individual researchers, consortia, and core facilities. The protocols that were evaluated span from droplet-based methods (10x Genomics) to split-and-pool (Evercode WT Mini), plate-based (SS3, FS, PW, VS, Sort-seq) and microwell devices (HIVE). This is a well-chosen selection, featuring the most used technologies in the scRNA-seq field. Their performance was evaluated across a range of different quality control parameters. Here as well the most important statistics were considered.

The conclusions that are drawn after this benchmarking are sound and reflect what several other papers have already reported in smaller-scale comparisons or when benchmarking their methods against competitors.

However, the methods need a thorough makeover (in some cases analyses are completely wrong, in my opinion), which raises the question whether the conclusions the Authors are drawing will remain valid afterwards.

COMMENTS - MAIN TEXT

Lines 347-348: “ (…) progressively yielding less reads on the 3’ end of the genes”.

I think it should be “5’ end of the gene”.

Lines 349-350: “10X and SORT-seq also yielded reads all over the transcript, which 349 previously has been attributed to semi-random binding of internal polyA repeats”.

The same phenomenon could be observed for Evercode WT Mini and HIVE/HIVE CLX but the Authors do not mention that in the manuscript. Please comment.

Lines 379-380: “The Plexwell and FLASH-seq methods are derived from Smart-seq3, and group together here”.

This is not correct, please change. FLASH-seq and Smart-seq3 were developed independently and both are based on Smart-seq2. Plexwell is (most likely) the commercial version of FLASH-seq, as protocols are the same up to the pre-amplified cDNA.

Interestingly, on lines 502-504, the Authors reported the “genealogy” of Smart-seq methods correctly.

COMMENTS – SUPPLEMENTARY FILE (METHODS)

Lines 149-150: “The FLASH-seq workflow[4] is derived from the Smart-seq3 workflow [2] and in most parts either identical or similar”. This is not correct, please change (see comment above).

Lines 193-210: although this is not going to affect the final results, it should be mentioned that tagmentation and enrichment PCR protocols are not those used In the original FLASH-seq paper.

Line 314: “Computational methods

Please include detailed information regarding which features are counted (and how) for every method (lncRNA, protein coding, exon, intron, etc.). This should be harmonised for all methods. Please also provide a link to the code used.

Line 325: Why using UMIdedup for a method without UMI? This should be set to “NoDedup”.

Line 326-327: “Reads per cell were counted with a custom script by parsing the SAM file”.

Ambiguous. Do the Authors mean counting sequencing depth or number of reads associated with each gene? If the latter is correct, please provide more information about it. Does it count exon, intron, both, … ? Which features are considered (protein coding, lncRNA, …)? Please provide a rationale for not using standard tools, such as featureCounts, STARsolo --quantMode or Kallisto/salmon.

Line 332: Cutadapt is not appropriate for this, as it does not handle well sparse mismatches. One could have used umi_tools instead or the zUMI pipeline developed by the Authors of Smart-seq3.

Lines 337-338: “The umis were trimmed of the umi reads with cutadapt and a minimum overlap of 8 to the full umis”.

Unclear, please rephrase.

Line 338: “Reads without umis were discarded”.

Reads without UMI are internal reads and should not be discarded rather set aside as they don’t contain the same information. As the Authors stated above (i.e. that the reads are separated based on the UMI presence), it is odd to add another separation step here.

Line 338-339: “A non-degenerated umi fastq file was generated based on the umis via a custom script».

Please provide more information and rational for not using standard pipelines.

It is unclear what the Authors mean by “non-degenerated UMIs” at this stage and what this processing step means.

Line 381: section “Normalization

This entire section is difficult to follow and, most importantly, not correct, both in terms of the rationale behind it as well as from the methodological standpoint.

Typically, normalisation for the sequencing depth is achieved through a size factor adjustment (non-UMI methods) or by collapsing the duplicated reads (=UMI). It should be performed at the cell level in such complex dataset and not bulk level.

Line 387: “The amount of passing cells was then multiplied by 20.000 and divided by the total number of clusters”.

Why was this number chosen? Please justify.

Lines 393-401: This procedure makes little sense and can lead to substantial artefacts. The number of clusters has nothing to do with the depth normalisation and this approach lacks the log-based rescaling of outlier counts.

Moreover, it does not take into account the vastly different number of cells analysed. When taking 10,000 reads from plate-based methods or 10,000 reads from 10x Genomics, the Authors are sampling a greatly different number of cells. This can therefore lead to a significant difference in gene diversity and could be the reason why their analysis does not saturate for most methods.

One approach could be to down-sample each cell to 20,000 reads and resample >1000 times, 1 to 100 cells per condition and then compare the number of detected genes.

Reviewer 2 Report

Comments and Suggestions for Authors

The manuscript explores the performance of various single-cell sequencing techniques, including Smart-seq3, 10X, HIVE, Evercode, VASA-seq, FLASH-seq, and Plexwell. Overall, the manuscript is well-structured and contributes significantly to the scientific community's understanding of these methods. However, I have a few comments and suggestions for improvement:

1.The manuscript states that "In humans, Plexwell, HIVE, and VASA-seq take spots afterward at different sequencing depths" (Lines 320-321). However, from Figure 7, it appears that Smartseq3 body and Vasa-seq are the two least effective methods in humans. Additionally, the HIVE curve seems to be positioned in the middle of the curves for all methods. To address this discrepancy, the authors may consider revisiting their interpretation and providing clearer distinctions in Figure 7 by using more varied colors for better differentiation.

2.The author calculates the relative coverage of each gene per exon for all methods. It is recommended to include a supplementary table to present these results systematically, enhancing accessibility and facilitating readers' understanding.

3.In Figure 9, it is suggested to include arrows in Panel B to highlight the enlarged bulk RNAseq points for better clarity and emphasis.

4.In line 375, "Figure 9A" should be corrected to "Figure 9B".

5.Clarify if "TrueSeq Stranded mRNA" in Figure 10 refers to 'bulk RNAseq' and provide an explanation in the manuscript for its recommendation. Additionally, include a legend in Figure 10 to enhance reader understanding.

6.In section "4.3 Multiplets", the authors should provide more details on how they inferred the multiplet rate.
